

# Effectiveness of water-saving technologies during early stages of restoration of endemic *Opuntia* cacti in the Galápagos Islands, Ecuador

Patricia Isabela Tapia[1,2,*], Luka Negoita[2,*], James P. Gibbs[3] and Patricia Jaramillo[2,4]

[1] School of Natural and Environmental Sciences, Newcastle University, Newcastle Upon Tyne, Tyne and Wear, United Kingdom
[2] Charles Darwin Research Station, Charles Darwin Foundation, Santa Cruz, Galápagos, Ecuador
[3] Department of Environmental and Forest Biology, State University of New York, Syracuse, NY, United States of America
[4] Facultad de Ciencias, Universidad de Málaga, Málaga, Spain
[*] These authors contributed equally to this work.

Corresponding author
Patricia Jaramillo,
patricia.jaramillo@fcdarwin.org.ec

## ABSTRACT

Restoration of keystone species is a primary strategy used to combat biodiversity loss and recover ecological services. This is particularly true for oceanic islands, which despite their small land mass, host a large fraction of the planet's imperiled species. The endemic *Opuntia* spp. cacti are one example and a major focus for restoration in the Galápagos archipelago, Ecuador. These cacti are keystone species that support much of the unique vertebrate animal community in arid zones, yet human activities have substantially reduced *Opuntia* populations. Extreme aridity poses an obstacle for quickly restoring *Opuntia* populations though water-saving technologies may provide a solution. The aim of this study was to evaluate current restoration efforts and the utility of two water-saving technologies as tools for the early stages of restoring *Opuntia* populations in the Galápagos archipelago. We planted 1,425 seedlings between 2013 and 2018, of which 66% had survived by the end of 2018. Compared with no-technology controls, seedlings planted with Groasis Waterboxx® water-saving technology (polypropylene trays with water reservoir and protective refuge for germinants) had a greater rate of survival in their first two-years of growth on one island (Plaza Sur) and greater growth rate on four islands whereas the "Cocoon" water-saving technology (similar technology but made of biodegradable fiber) did not affect growth and actually reduced seedling survival. Survival and growth rate were also influenced by vegetation zone, elevation, and precipitation in ways largely contingent on island. Overall, our findings suggest that water-saving technologies are not always universally applicable but can substantially increase the survival and growth rate of seedlings in certain conditions, providing in some circumstances a useful tool for improving restoration outcomes for rare plants of arid ecosystems.

## INTRODUCTION

The restoration of previously abundant keystone species is one way to combat loss of biodiversity and ecological services (*Grime, 1998*). This is particularly true on oceanic islands, which comprise little of the planet's land mass yet host a disproportionate amount of its imperiled species (*Myers et al., 2000*; *Campbell & Donlan, 2005*). The Galápagos archipelago is a case in point: its land area is minimal (8,006 km$^2$) yet it hosts a remarkable array of endemic life forms with as many as 60% of its 168 endemic plant species now threatened with extinction (*Black, 1973*; *Tye, 2007*). Active restoration programs are underway throughout the archipelago. For example, Project Isabela (1997–2006), was the world's largest restoration effort at the time and dedicated to eradicating introduced mammal herbivores on multiple islands in the archipelago (*Cruz et al., 2009*; *Carrion et al., 2011*).

The *Opuntia* spp. cacti (prickly pear cactus) are a major focus for restoration in the Galápagos archipelago, Ecuador, which hosts six endemic species, with 14 total taxa when including varieties. Human impact in the Galápagos archipelago has steadily increased over the last 200 years (*Jaramillo, 1998*), resulting in declines of *Opuntia* populations on these islands (*Snell, Snell & Stone, 1994*). Several factors have been attributed as the primary threats to Opuntias including herbivory by introduced mammals (*Grant & Grant, 1989*), extinction of keystone predators that once regulated numbers of cactivores (*Sulloway & Noonan, 2015*), and the increased intensity of El Niño events likely driven by climate change (*Snell, Snell & Stone, 1994*; *Hicks & Mauchamp, 1996*). *Opuntia* cacti provide many ecosystem services for other native and endemic species (*Grant & Grant, 1981*; *Hicks & Mauchamp, 1995*; *Hicks & Mauchamp, 1996*; *Gibbs, Marquez & Sterling, 2008*). Examples include Galápagos giant tortoises and land iguanas that depend on *Opuntia* cacti as a food source while also contributing to *Opuntia* regeneration through seed dispersal (*Hamann, 1993*; *Snell, Snell & Stone, 1994*; *Gibbs, Marquez & Sterling, 2008*; *Gibbs, Sterling & Zabala, 2010*; *Jaramillo, Tapia & Gibbs, 2018*). Efforts are being made to protect and restore populations of these imperiled cacti (*Hicks & Mauchamp, 1996*) but it is not clear which factors most control *Opuntia* populations (*Sulloway & Noonan, 2015*). *Opuntia* declines on Plaza Sur Island, for example, are especially pronounced (60% reduction since 1957) despite the eradication of introduced goats since the populations are likely too low to successfully regenerate in the presence of native herbivory (*Grant & Grant, 1989*; *Snell, Snell & Stone, 1994*; *Sulloway & Noonan, 2015*).

Severe aridity poses an obstacle for restoring plant communities over much of Galápagos due to the inherently slower growth and low germination of plants growing in these conditions, including xerophytes such as *Opuntia* cacti (*Hicks & Mauchamp, 1996*). The lowland zones of the archipelago, where Opuntias are most common and historically abundant (e.g., *Snell, Snell & Stone, 1994*; *Hicks & Mauchamp, 1996*; *Browne et al., 2003*), can receive less than 10 cm rainfall annually (*Trueman & D'Ozouville, 2010*). Though these conditions are normal, they increase the time it would take for small populations of Opuntias to return to historic sizes (*Grant & Grant, 1989*; *Helsen et al., 2009*). Rapid restoration through active planting of these species is critical for reducing the risk of

extinction until their threats are better understood and before other threats such as invasive plant species make it more difficult or impossible for Opuntias to naturally regenerate (*Mauchamp et al., 1998*; *Helsen et al., 2009*). ''Water-saving'' technologies are tools that may help increase survival and growth of planted cactus seedlings while reducing the need for manual watering and speeding the restoration process (*Kulkarni, 2011*; *Hoff, 2014*; *Jaramillo et al., 2014*; *Jaramillo, 2015*; *Jaramillo et al., 2015*; *Faruqi et al., 2018*; *Jaramillo, Tapia & Gibbs, 2018*; *Peyrusson, 2018*; *Peyrusson, 2018*). The Groasis Waterboxx® (Groasis) and biodegradable Cocoon system are two relatively inexpensive water-saving technologies that can be easily implemented during the planting process (Appendix S1). These technologies function by holding water in basins that surround the young plant and feed water to the soil at a slow but constant rate through capillary action via a short length of rope that connects the basin to the soil. Aside from the physical design differences that influence where the plant is relative to soil surface and biodegradability, the main difference in these technologies is that Groasis actively collects dew and rainwater, while the Cocoon technology is only filled with water once at the time of planting (Appendix S1). Although these particular technologies show much promise through anecdotal evidence and reports, there remains a dearth in formal scientific studies evaluating their efficacy (but see *Liu, Li & Ren, 2014*). Therefore, the objective of the current study was to evaluate current restoration efforts and test the utility of two water-saving technologies as tools for restoring *Opuntia* populations in the Galápagos archipelago. Through this objective we hope to better understand the utility of water-saving technologies for restoring these and other keystone plant species in arid island ecosystems throughout the world.

## MATERIALS & METHODS

### Study area, focal species, and water-saving technologies

The Galápagos archipelago is located in the Pacific Ocean, about 1,000 km west of the coast of mainland Ecuador (1°39′N, 92°0′W to 1°26′S, 89°14′W, WGS 84, Fig. 1) (*Dirección del Parque Nacional Galápagos, 2014*). Our study focused on measuring the utility of water-saving technologies for enhancing cactus growth and survival of four endemic *Opuntia* taxa within the archipelago: *Opuntia echios* var. *echios* Howell, *Opuntia echios* var. *gigantea* Howell, *Opuntia megasperma* var. *megasperma* Howell, and *Opuntia megasperma* var. *orientalis* Howell (*Hicks & Mauchamp, 1996*). We evaluated two technologies: Groasis Waterboxx® (Groasis), a protective polypropylene box that collects rainwater that it provides to the plant (*Hoff, 2014*); and the Cocoon system, a 99% biodegradable box that contains and provides water to the plant similar to Groasis, but Cocoon is only filled with water at the time of planting (*Land Life Company, 2015*; *Faruqi et al., 2018*; Appendix S1). These water-saving technologies have been proposed as a tool to assist plant restoration of *Opuntia* taxa via ''Galápagos Verde 2050'' (GV2050), a project started by the Charles Darwin Foundation in 2013 with the mission of restoring degraded ecosystems and aiding with sustainable agriculture in the Galápagos archipelago (*Jaramillo et al., 2014*; *Jaramillo et al., 2015*; *Jaramillo, Tapia & Gibbs, 2017*). GV2050 seeks to restore ecosystems by using
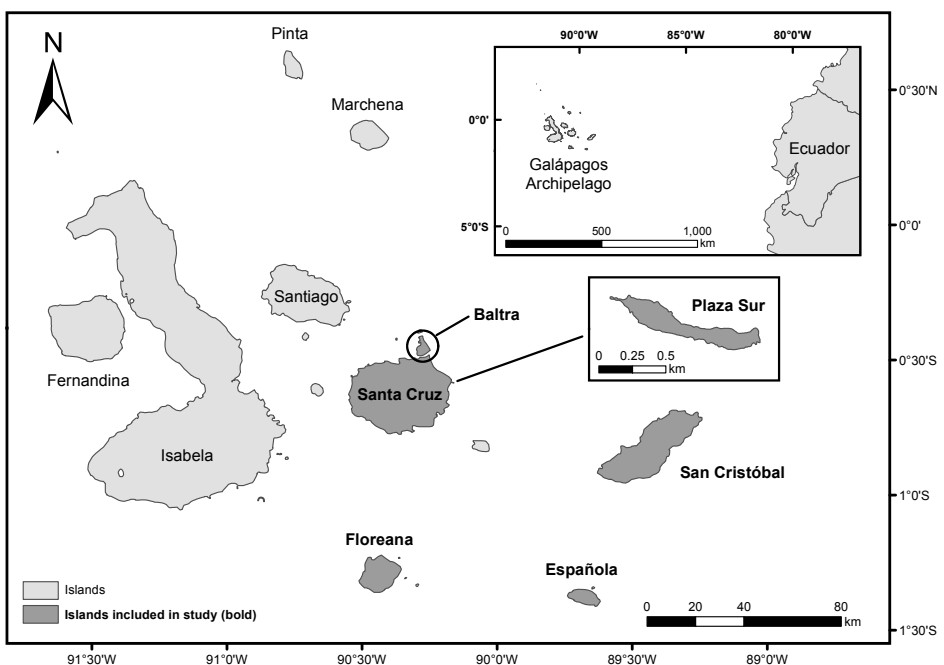

**Figure 1** **Map of the Galápagos Islands, Ecuador.** Islands included in the current study are darkened and labeled in bold.

a data-informed experimental approach for understanding the best conditions, methods, and tools for successful plantings of native and endemic species (*Jaramillo et al., 2015*).

## Planting and data collection

A total of 1,425 cacti (1,137 *Opuntia echios* var. *echios*, 68 *Opuntia echios* var. *gigantea*, 24 *Opuntia megasperma* var. *megasperma*, and 196 *Opuntia megasperma* var. *orientalis*) were planted on six islands (Baltra, Española, Floreana, Plaza Sur, San Cristóbal, and Santa Cruz) between 2013 and 2018 (Table 1). Permission to plant Opuntias within protected sites on these islands was granted by the Dirección del Parque Nacional Galápagos (DPNG) through permit number PC-11-19 (Table 2). To evaluate the factors most important for successful *Opuntia* restoration, data were used only from Opuntias that were grown from seed and planted using either Groasis, Cocoon, or control (no technology) treatments on Floreana, Santa Cruz, Baltra, and Plaza Sur islands yielding a sample of 1,029 *Opuntia* individuals of three taxa (Table 1).

Planting sites on each island were selected based on locations where historic *Opuntia* populations were known to have thrived but are now in decline (*Hicks & Mauchamp, 1996*; *Sulloway et al., 2013*; *Sulloway & Noonan, 2015*; Table 2). For example, since 1957 the *Opuntia* population on Plaza Sur Island has had an overall mortality of more than 60% and at Cerro Dragon on Santa Cruz Island there has been an overall loss of 78% (*Sulloway & Noonan, 2015*). Seedlings were planted from seeds collected in each respective planting location using standardized seed collection and stratification techniques and

**Table 1** **Total number of *Opuntia* spp. individuals planted by island by Galápagos Verde 2050 (2013–2018).** Numbers in parentheses '()' are the number of individuals used in the current study analysis (Figs. 3 & 4).

| Species | Baltra | Española | Floreana | Plaza Sur | San Cristóbal | Santa Cruz |
|---|---|---|---|---|---|---|
| *Opuntia echios* var. *echios* | 400 (349) | – | – | 737 (601) | – | – |
| *Opuntia echios* var. *gigantea* | – | – | – | – | – | 68 (60) |
| *Opuntia megasperma* var. *megasperma* | – | – | 20 (19) | – | 4 (0) | – |
| *Opuntia megasperma* var. *orientalis* | – | 196 (0) | – | – | – | – |

**Table 2** **List of all sites of Galápagos Verde 2050 *Opuntia* spp. restoration and number of *Opuntia* spp. individuals planted (2013–2018).** Numbers in parentheses '()' represent the percent of individuals that have survived through 2018.

| Island | Site Name | # Planted | UTM East[a] | UTM North[a] |
|---|---|---|---|---|
| Baltra (70%) | Antiguo basurero | 158 (69%) | 804668 | 9950436 |
| | Casa de piedra | 125 (74%) | 802460 | 9948203 |
| | Jardín ecológico Aeropuerto | 1 (100%) | 804100 | 9950795 |
| | Parque Eólico | 116 (68%) | 803992 | 9950909 |
| Española (79%) | Las Tunas | 196 (79%) | 199759[b] | 9849118[b] |
| Floreana (40%) | Botadero de basura | 3 (33%) | 781054 | 9858587 |
| | Cementerio | 7 (29%) | 780322 | 9858645 |
| | Escuela Amazonas | 5 (40%) | 779594 | 9858865 |
| | Gobierno Parroquial Floreana | 1 (0%) | 779530 | 9859029 |
| | Oficina Técnica Parque Nacional Galápagos | 4 (75%) | 779531 | 9859244 |
| Plaza Sur (61%) | Centro | 254 (62%) | 815800 | 9935365 |
| | Los Lobos Este | 253 (47%) | 815936 | 9935354 |
| | Oeste Cerro Colorado | 230 (76%) | 815304 | 9935602 |
| San Cristóbal (100%) | CA Jacinto Gordillo | 4 (100%) | 209711[b] | 9900150[b] |
| Santa Cruz (65%) | Colegio Nacional Galápagos | 2 (50%) | 798782 | 9918296 |
| | Espacio Verde ABG | 8 (88%) | 797864 | 9918887 |
| | Fundación Charles Darwin | 51 (67%) | 800106 | 9917856 |
| | Oficina Técnica Parque Nacional Galápagos | 7 (29%) | 799811 | 9917994 |

**Notes.**
[a]UTM Zone = 15M, datum = WGS84
[b]UTM Zone = 16M

grown for one year at the Charles Darwin Research Station, Santa Cruz Island, before transferring to each site on each island (*Jaramillo, Tapia & Gibbs, 2017*, Table 2). Each seedling was randomly assigned a treatment of either control (no technology), Groasis, or Cocoon, ensuring an adequate sample of replicates within each treatment and site. The number of controls was maintained at one control for every five technology treatment replicates. A greater proportion of Groasis replicates were used because the overarching goal of this work is to successfully restore populations of Opuntias and current anecdotal evidence and observations suggest this technology provides the greater benefit for achieving this. The uneven design does not impact our analyses or interpretation of results since we ensured a relatively adequate number of controls within each island. In total, 823 Groasis, 38 Cocoons, and 168 controls were used in the analysis. Planting locations for

each seedling were determined haphazardly in the field at each site using the basic criteria that a seedling could be physically planted while not being in direct competition with other plants (i.e., the substrate was soil rather than rock and free of overarching vegetation that would shade the seedling). Truly random selection of specific planting locations was impractical due to the large heterogeneity of exposed bedrock and competing vegetation, so locations were often chosen opportunistically. Though planting locations were not random, treatment assignment was random and thus our methodology does not interfere with our primary goal of evaluating the use of water-saving technologies. Plantings were conducted according to established methods for installing Groasis, Cocoon, and controls (*Miranda, Riganti & Tarrés, 1987*; *Hoff, 2014*; *Land Life Company, 2015*). Wire fences were secured and maintained around each individual planting on Plaza Sur, Baltra, and Española islands to prevent herbivory from land iguanas or giant tortoises present on those islands but absent from other islands where Opuntias were planted. Planting site co-variates were recorded at time of planting: elevation, soil type (rocky-sand, rocky-clay, rich-clay, rich, sandy, and clay), vegetation zone (arid, littoral, and transitional; *Johnson & Raven, 1973*), and treatment (control, Groasis, and Cocoon). Growth (vegetative height) and qualitative plant state ("good", "regular", "poor", and "dead") were noted during each repeated visit approximately every six weeks following planting. Aside from "dead" which was non-arbitrary and easy to identify, the other states were based on the subjective relative appearance of the plant (i.e., degree of desiccation or browning of cladodes). Though these assignments were not objective, they were not used for our analysis and simply provide a quick way to gauge the relative state of the plants.

Two-year survival and growth rate of seedlings were used to evaluate restoration success (*Menendez & Jaramillo, 2015*). Two-year survival was quantified as whether or not a seedling survived for at least two years after planting—the period of greatest mortality risk (we found 79% survival in the first year and 86% survival in the second year, compared with 99% survival in the third year). For this analysis, only plants that had the potential to grow and survive for two years were included. However, while the analysis was based on seedlings planted up until 2019, additional monitoring data from those plants until September 2019 allowed us to increase the sample of plants for which we could model two-year survival. Relative growth rate was calculated based on the vegetative height of each seedling over time. Whereas survival is the primary metric for establishing success of population restoration, growth rate can indicate the speed of ecosystem recovery due to the rate of increase in the biomass of a keystone species (*Grime, 1998*), and may also indicate the time to reproductive maturity in Opuntias (*Racine & Downhower, 1974*). An additional environmental covariate of total precipitation across the six months following planting was compiled based on available climate data from 2013 to 2019 (*Trueman & D'Ozouville, 2010*; *Charles Darwin Foundation, 2018*).

## Data analysis

All statistical analyses were conducted using the R statistical software package v3.3.3 (*R Core Team, 2017*). To test the overall effect of water-saving technologies on the restoration

of *Opuntia* cacti, a model comparison approach was implemented using fixed- and mixed-effects regression models of the form:

## 2-year survival logistic fixed-effect model

$2YearSurvival = \alpha + \beta_1 \times treatment + \beta_2 \times 6MonthPrecip + \beta_3 \times Zone + \beta_4 \times elevation + \beta_5 \times island$

## Relative growth rate linear mixed-effect model

$log(RGR) = \alpha + \beta_1 \times treatment + \beta_2 \times 6MonthPrecip + \beta_3 \times SoilType + \beta_4 \times Zone + \beta_5 \times elevation + \beta_6 \times PlantAge + \beta_6 \times island + N(0, \sigma^2_{PlantID})$

The growth rate model is a general linear mixed-effects regression fit using the 'lme4' package (*Bates et al., 2015*). Relative growth rate (RGR) was calculated as the relative rate of increase in height over time and was log-transformed to meet assumptions of normality. Growth rates of zero were excluded from this analysis to maintain normality. Plant age was included in the model to account for the fact that RGR changes as seedlings get older. Plant ID is included as a random effect. Random effects account for within-group correlation that results from non-independent data points (*Pinheiro & Bates, 2000*). For example, our growth data are based on repeated measures of each individual plant, which means that growth measurements are not independent within an individual plant. The random effect for Plant ID allows us to include all observations in our analysis by accounting for this non-independence. The two-year survival model tested the overall survival of each seedling two years after planting and was fit using a generalized linear model function with a binomial family logit function in the 'base' package (*R Core Team, 2017*). Because only one data point was available for each plant, the lower sample size required a simpler model in which soil type was removed in order to allow the model to converge successfully and no random effects were necessary. These models were then compared to null models using the likelihood-ratio to test for the effect of treatment on growth rate and survival. Null models were the same as the models listed above except for the exclusion of technology treatment. A significant difference between the two models indicates that the variable that was excluded (i.e., treatment) is a significantly important predictor.

We examined the relative effect of each variable within the growth rate and survival models to assess the relative importance of technologies as well as other environmental factors such as soil type and elevation. All continuous variables in our models were standardized by subtracting the mean and dividing by two times the standard deviation in order to relativize the effect of each variable coefficient on growth rate and two-year survival (*Gelman, 2008*). Confidence intervals (95%) for each coefficient in each full model were then generated through the "profile" method (*Stryhn & Christensen, 2003*) and plotted for visual comparison. *P*-values were generated for each coefficient in the logistic regression based on the Wald statistic. For the mixed effect growth rate model, *P*-values were generated using the Satterthwaite method in the 'lmerTest' package in R

(*Kuznetsova, Brockhoff & Christensen, 2017*). *P*-values generated from mixed-effect models are not always accurate, but we include these values for the sake of highlighting the degree to which variables differ in their relative importance. Furthermore, all significance values generated in this way were consistent with confidence interval results. Coefficients for logistic models were back-transformed to odds ratio by exponentiating and subtracting one. In this way the coefficient values can be interpreted as the proportional effect of each variable on increasing (or decreasing if negative) the probability of two-year survival. Each model was fit using data from all four islands included in the analysis (Baltra, Floreana, Santa Cruz, and Plaza Sur), but due to high control treatment mortality on Plaza Sur, the models were also tested using data that *excluded* Plaza Sur as well as using data *exclusively from* Plaza Sur. Continuous variables were standardized within each of these three analyses. When testing with data exclusively from Plaza Sur, "island" was removed from the models and treatment type consisted of only Groasis and controls because no Cocoons were used on Plaza Sur. Finally, the current state of all planted individuals included in the analysis (up through 2018) was plotted as stacked bar plots to visualize rates of survival between islands and treatments.

# RESULTS

## General outcomes

Of the 1,425 *Opuntia* spp. individuals planted between 2013 and 2018, (most plantings were made in 2015 and 2016, Fig. 2), 943 Opuntias remained alive by the end of 2018 (66% overall survival, Fig. 2). Of those individuals planted at least three years prior to 2019, *Opuntia* mortality three years after planting fell to 1% and overall survival leveled at 67%. On Plaza Sur, 737 *Opuntia* individuals were planted between 2015 and 2018 with 452 survivors by the end of 2018 (an increase of 106% from the last recorded population estimates of 426 in 2014 (*Jaramillo, Tapia & Gibbs, 2017*)). Survival of seedling plantings on Plaza Sur was 26.8% ($n = 82$) for controls and 62.2% ($n = 519$) for Groasis (Fig. 3A). Survival of seedling plantings on Floreana was 66.7% ($n = 3$) for controls and 31.2% ($n = 16$) for Groasis (Fig. 3B). Survival of seedling plantings on Baltra was 79.7% ($n = 74$) for controls, 45% ($n = 20$) for Cocoon, and 65.5% ($n = 255$) for Groasis (Fig. 3C). Survival of seedlings planted on Santa Cruz was 77.8% ($n = 9$) for controls, 27.8% ($n = 18$) for Cocoon, and 72.7% ($n = 33$) for Groasis (Fig. 3D).

## Outcomes across all islands

Treatment type (Groasis, Cocoon, or Control) was associated with growth rate ($\chi^2$ (2) = 54.54, $P < 0.001$) and two-year survival rate of *Opuntia* seedlings ($\chi^2$ (2) = 41.53, $P < 0.001$). In the two-year survival logistic regression, elevation (1.88, $P = 0.001$) and littoral zone (13.72, $P < 0.001$) had odds ratios with confidence intervals that did not overlap zero (Fig. 4A). Groasis technology had a positive odds ratio of 1.28 ($P < 0.001$), while Cocoon had a negative odds ratio of −0.89 ($P < 0.001$) (Fig. 4A). In the growth rate regression, littoral zone (0.48, $P < 0.001$), plant age (−0.51, $P < 0.001$), rocky-sand soil (−0.3, $P = 0.026$), and six-month precipitation (0.23, $P = 0.033$) all had effect sizes with confidence intervals that did not overlap zero (Fig. 4B). Groasis technology had a
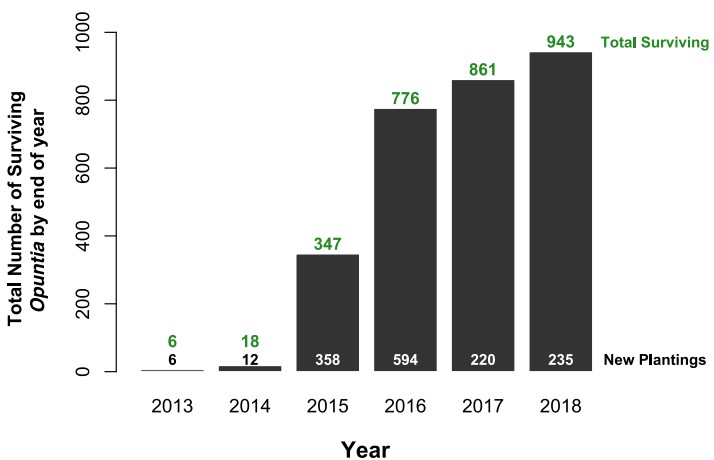

**Figure 2** Total *Opuntia* spp. restoration from 2013 to 2018 across Baltra, Española, Floreana, Plaza Sur, San Cristóbal, and Santa Cruz islands. Values above bars indicate total surviving individuals by the end of each year (*y*-axis values). Values at the bottom indicate the total number of individuals planted each year.

positive effect size with a coefficient of 0.52 ($P < 0.001$), while Cocoon had an insignificant coefficient ($P = 0.179$) (Fig. 4B).

## Outcomes on plaza sur island only

On Plaza Sur Island, treatment type (Groasis or Control) was associated with growth rate of *Opuntia* species ($\chi^2$ (1) $= 18.92$, $P = 0.001$) and two-year survival rate of *Opuntia* seedlings ($\chi^2$ (1) $= 23.44$, $P < 0.001$). In the two-year survival logistic regression, littoral zone (379.63, $P < 0.001$), elevation (1.54, $P < 0.001$), and six-month precipitation ($-0.67$, $P < 0.001$) had odds ratios with confidence intervals that did not overlap zero (Fig. 4C). Groasis technology had a positive odds ratio of 3.7 ($P < 0.001$) (Fig. 4C). In the growth rate regression, littoral zone (0.49, $P < 0.001$), plant age ($-0.26$, $P < 0.001$), six-month precipitation ($-0.23$, $P = 0.001$), and elevation (0.17, $P = 0.012$) all had effect sizes with confidence intervals that did not overlap zero (Fig. 4D). Groasis technology had a positive effect size with a coefficient of 0.46 ($P < 0.001$) (Fig. 4D).

## Outcomes on all islands excluding Plaza Sur

Treatment type (Groasis, Cocoon, or Control) was associated with growth rate of *Opuntia* species ($\chi^2$ (2) $= 17.8$, $P < 0.001$), but not with two-year survival rate of *Opuntia* seedlings ($\chi^2$ (2) $= 1.85$, $P = 0.397$). In the two-year survival logistic regression, transition zone (-0.99, $P < 0.001$) and littoral zone ($-0.77$, $P = 0.013$) had negative odds ratios with confidence intervals that did not overlap zero (Fig. 4E). Both Groasis and Cocoon technologies had insignificant negative odds ratios of ($P = 0.236$) and ($P = 0.305$) respectively (Fig. 4E). In the growth rate regression, plant age ($-0.83$, $P < 0.001$), six-month precipitation (0.52, $P < 0.001$), and rocky-clay soil ($-0.23$, $P = 0.032$) had effect sizes with confidence intervals that did not overlap zero (Fig. 4F). Groasis technology had

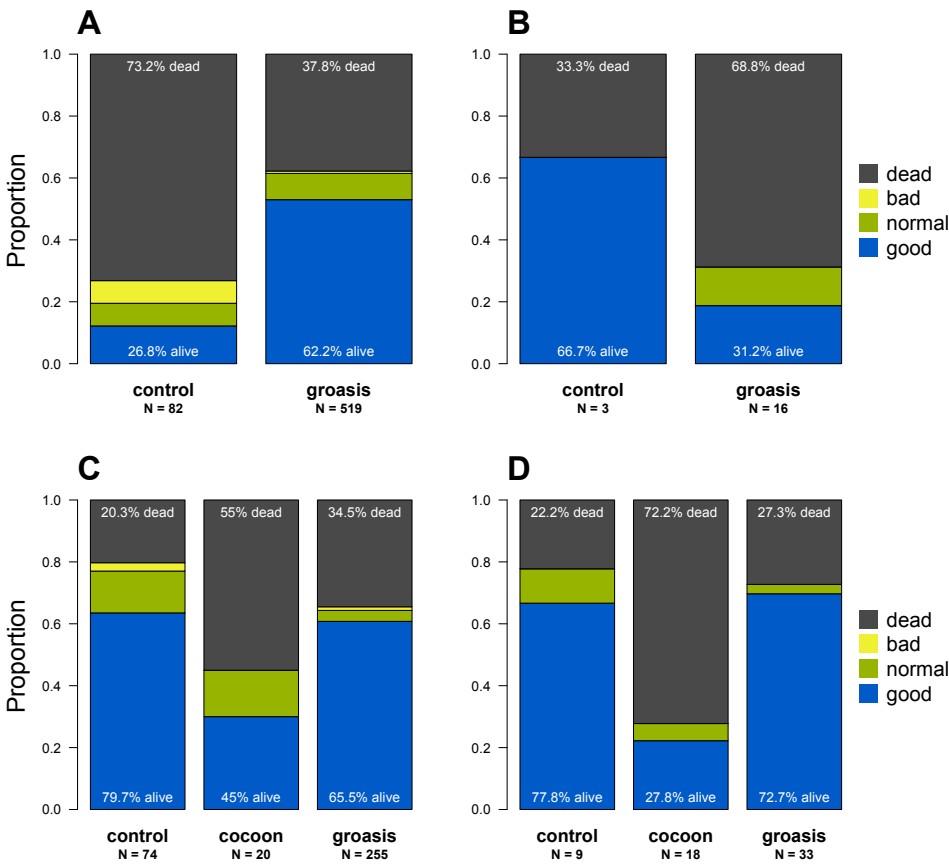

**Figure 3  State of each planted *Opuntia* individual by the end of 2018 within each island.** (A) Plaza Sur; (B) Floreana; (C) Baltra; (D) Santa Cruz. "N" indicates the total number of individuals within each treatment on each island. Plant state categories ("good", "regular", "poor", and "dead") refer to the subjective observation of the physical state of the plant. "Good" plants are fully green with no signs of desiccation or browning in the cladodes, while "poor" plants appear desiccated and browning, and likely to die soon. The figure is based on the last noted observation of each plant at the end of 2018 and based on only those data used in the analysis.

a positive effect size with a coefficient of 0.4 ($P < 0.001$), while cocoon had an insignificant coefficient ($P = 0.261$) (Fig. 4F).

# DISCUSSION

Water-saving technologies enhanced survival and growth of *Opuntia* plantings, but benefits of these technologies were highly contingent upon planting environment. For example, Groasis technology was effective at increasing growth rate across islands overall, but was only effective at aiding survival on Plaza Sur Island where Groasis increased the probability of two-year survival of seedlings more than three-fold (370%) (Fig. 4). Cocoon technology, however, provided no improvement in growth rate and actually reduced probability of two-year survival of seedlings by 89% overall (Fig. 4). Although still in its early stages with all planted Opuntias yet to reach maturity, our restoration efforts have increased the

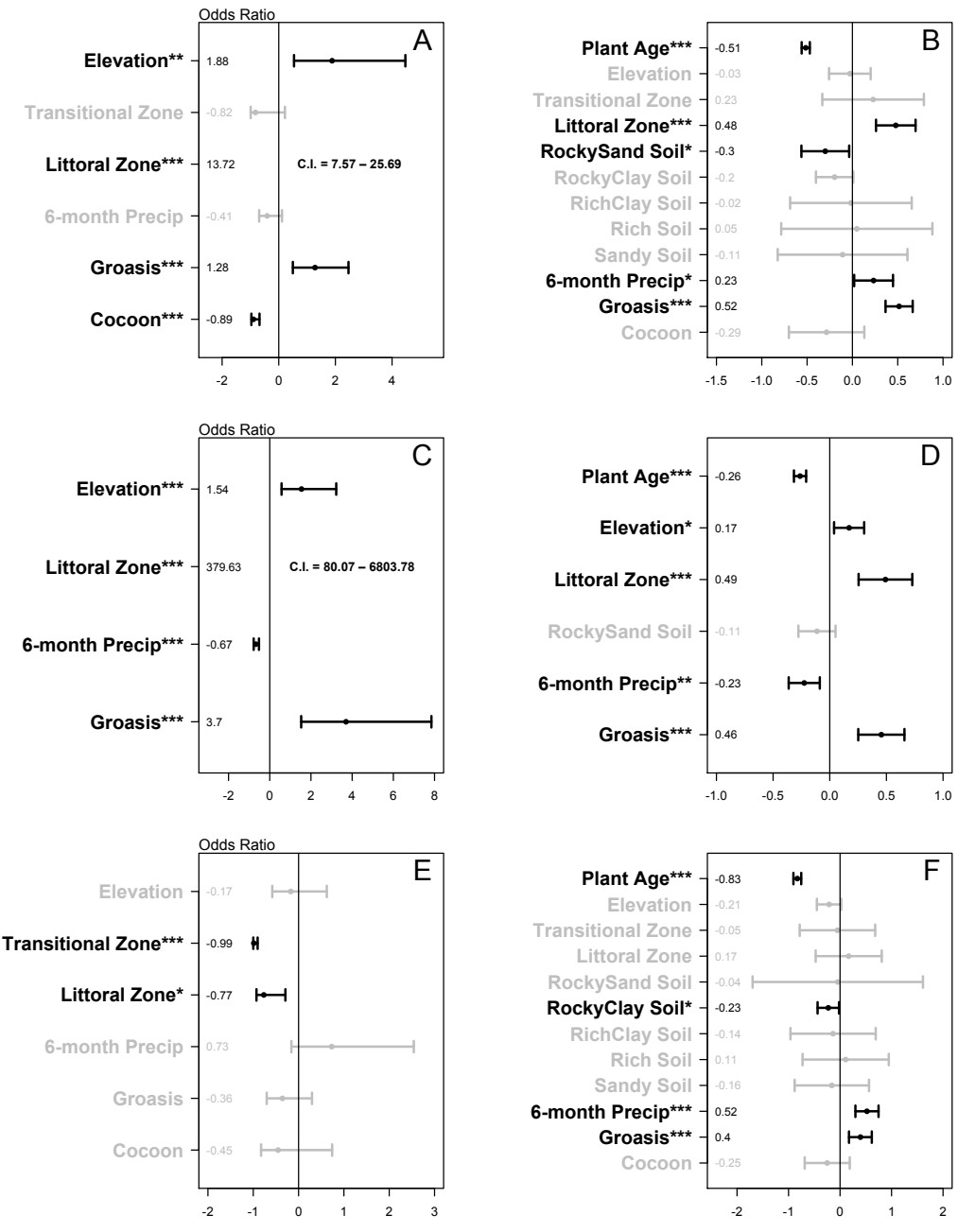

**Figure 4** **Plots of the relative effect of variable parameters on two-year survival and growth rate of planted *Opuntia* individuals.** (A) all islands two-year survival; (B) all islands growth rate; (C) Plaza Sur island two-year survival; (D) Plaza Sur island growth rate; (E) all islands excluding Plaza Sur two-year survival; and (F) all islands excluding Plaza Sur growth rate. Each point represents coefficient estimate +/- 95% confidence intervals. *P*-values are generated based on the Satterthwaite method for growth rate models and the Wald statistic for survival models (* $P < 0.05$, ** $P < 0.01$, *** $P < 0.001$). Values for two-year survival models are converted to odds ratio by exponentiating coefficients and subtracting one. Analyses are based on data from Baltra, Floreana, Plaza Sur, and Santa Cruz islands. Littoral zone values in (A) and (C) fall outside the scale of those boxes, so confidence intervals are presented as text.

population of *Opuntia* spp. in the Galápagos archipelago by 943 individuals (66% survival of 1,425 plantings), more than doubling the population of *Opuntia* cacti on Plaza Sur Island, from 426 to 878 in just four years (*Jaramillo, Tapia & Gibbs, 2017*).

These results emphasize the species- and site-specific contingencies of applying water-saving technologies for plant restorations. For example, Cocoon technology did not provide any advantage when planting Opuntias in the Galápagos archipelago. This is despite the fact that in other systems and with other species Cocoon has been shown to increase survival rates in planted trees from 0–20% to 75–95% (*Faruqi et al., 2018*). One possible explanation is that *Opuntia* cacti have a short initial rooting depth compared to other species (*Snyman, 2005*), and this may reduce access to the water available from the Cocoon (*Land Life Company, 2015*; Appendix 1). *Acacia macracantha*, for example, has much deeper roots and has had much greater success when planted with Cocoon technology in the Galápagos (GV2050, *unpublished data*).

Although Groasis technology helped increase growth rate of Opuntias overall, it had a clear, positive effect on the survival of Opuntias only on Plaza Sur Island. A likely factor contributing to this is that compared to other islands, the majority of Opuntias were planted on Plaza Sur preceding the greatest period of drought in the Galápagos over the last five years (Appendix 2; *Charles Darwin Foundation, 2018*). Despite fairly regular seasonal patterns of water availability in the Galápagos (*Snell & Rea, 1999*; *Restrepo et al., 2012*), there remains much variability, especially caused by El Niño events (*Trueman & D'Ozouville, 2010*). In this way Groasis may have the greatest advantage when ensuring water availability for Opuntias during periods of especially severe drought, and in particular for seedlings which rely on periods of greater moisture to germinate and survive (*Hicks & Mauchamp, 1996*). *Opuntia* cacti are typically more resistant to desiccation and water stress compared to other species that do not have physiological adaptations for surviving low-water desert conditions (*Racine & Downhower, 1974*; *Dubrovsky, North & Nobel, 1998*), and this may explain why Groasis was only effective for *Opuntia* cacti under extreme drought. These findings support the idea that water availability for Opuntias plays less of a role in survival than previously assumed (*Racine & Downhower, 1974*; *Coronel, 2002*; *Jaramillo, Tapia & Gibbs, 2018*). These findings do not negate the value of the Cocoon or Groasis technology for restoration overall, but rather presents the important observation that water-saving technologies such as Cocoon and Groasis should be considered on a case-by-case basis and tested with each species and in different environmental conditions before making expansive planting efforts. Groasis technology may provide a form of insurance for the unpredictability of extreme drought events and the benefits of using Groasis technology may in some cases outweigh the costs in the long run (e.g., ~22 USD per Groasis unit plus overhead and installation time (*Groasis®, 2019*)).

Site co-variates also affected *Opuntia* survival and growth. In particular, vegetation zone, elevation, and precipitation were important predictors of *Opuntia* survival and growth but as with water-saving technologies, these effects were highly contingent on island. Opuntias had a greater survival and growth rate in the littoral vegetation zone on Plaza Sur but had greater survival in the arid vegetation zone on other islands. This effect may be due to an interaction between environmental and biotic factors unique to Plaza Sur or other islands.
For example, Plaza Sur has especially high land iguana densities speculated to be due to the loss of its main predator from the island, the Galápagos hawk (*Sulloway & Noonan, 2015*). This high herbivore density may help keep invasive plant species in check on Plaza Sur—species that may otherwise shade out *Opuntia* seedlings on other islands (*Schofield, 1973*; *Hicks & Mauchamp, 1996*; *Hicks & Mauchamp, 2000*).

Surprisingly, the level of precipitation six-months after planting did not increase seedling survival, and actually decreased survival of seedlings planted on Plaza Sur. This finding contradicts conclusions from previous work by *Coronel (2002)* who found that precipitation during the six months following planting increased *Opuntia* survival. *Coronel (2002)*, however, found that the positive effect of rainfall following planting was mostly evident in Opuntias grown from vegetative cladodes rather than seeds as in the current analysis. Furthermore, most seedlings were planted on Plaza Sur at the start of a long period of drought so there was not as much variation in precipitation on Plaza Sur seedlings to fully test its effects. Elevation was only a significant predictor of survival and growth rate on Plaza Sur (Fig. 4). This may be in part because although littoral zone on Plaza Sur has a positive impact on survival and growth, seedlings that are too low in elevation are more exposed to ocean salt spray which can increase seedling mortality (*Boyce, 1954*). Soil type had only marginally significant effects on growth rate (Fig. 4), suggesting that, at least for Opuntias, substrate is of less importance for growth rate than factors such as vegetation zone or elevation. The effect of soil type on survival could not be tested with the current data due to limitations in sample size.

The observational aspects of our study have some inherent limits. Although it seems likely that extreme drought was the primary driver of control treatment seedling mortality on Plaza Sur, other effects cannot be ruled out. Plaza Sur is a small island (the smallest island by far of the four in this analysis: only 13 ha, with the next larger being Baltra at 2100 ha), which could increase the exposure of seedlings to salt spray, exposure to sea lion activity, as well as a suite of other effects associated with small islands (*Lomolino & Weiser, 2001*). It may also be that the high concentration of land iguanas and sea lions (P Jaramillo, pers. obs., 2018) has impacted the edaphic environment of the island through their excrement as can be common on seabird islands (*Rajakaruna et al., 2009*). Thus, the small area and low variation in elevation, precipitation, and vegetation zones associated with Plaza Sur plantings suggests that any significant effect of these factors within Plaza Sur be taken cautiously when generalizing to *Opuntia* restoration beyond this island. The experimental treatments of the study involving water-saving technologies, however, do suggest that extreme drought is the most probable hypothesis for the high control mortality on Plaza Sur. Another important caveat is that taxon effects are confounded with island effects. With one exception, each island had a particular species or variety of *Opuntia* (Table 1). It is possible that some of the island-based differences are actually due to slightly different environmental requirements of the *Opuntia* taxa used in this study.

In conclusion, this study underlines the importance of considering the specific circumstances and methodologies that affect successful restoration. Water-saving technologies such as the Groasis Waterboxx® and Cocoon are promising systems for restoring species in arid environments but should not be assumed to function equally well

in all environments and with all species. Even within one system, as in the current study, the benefits of Groasis vary tremendously and likely depend on the precipitation available following plantings. It is possible that species already adapted for low water conditions, such as cacti, have a much higher threshold of drought at which Groasis or other water-saving technologies provide a benefit. Future evaluations of these technologies should monitor precipitation to test whether there is a threshold level of drought where these technologies become more effective. In some cases and for some species there may be no threshold for effective use as with the Cocoon technology for Opuntias. Preliminary plantings coupled with extensive environmental and experimental data collection is essential before large-scale planting efforts are initiated with water-saving technologies. Our work restoring reproductive *Opuntia* populations is still in its early stages, but water-saving technologies may have a profound influence on how quickly we reach sustainable levels of reproductive *Opuntia* populations on these islands. Field observations and unpublished data suggest that Opuntias reach reproductive maturity at between 20 and 40 years of age, largely dependent on the island and particular taxon (W Tapia, J Gibbs, & F Sulloway, 2019, unpublished data). A conservative estimate based on current planting survivorship is that at least 60% of planted individuals (855) will reach reproductive maturity (this is based on our three-year survival rate of 67%, at which point yearly mortality fell to 1%).

Through our experimental evaluation of restoration methodologies, the Galápagos Verde 2050 project of the Charles Darwin Foundation presents a model for data-informed adaptive management and conservation. We hope this model may inspire other restoration efforts to adopt similar data-informed approaches. Continued monitoring and accounting for context-specific contingencies in restoration work is essential (*Cabin, 2007*) and future restoration efforts should continually adapt management protocols based on current results (*Parma & NCEAS Working Group on Population Management, 1998*).

## ACKNOWLEDGEMENTS

This is the work of the entire Galápagos Verde 2050 project team of the Charles Darwin Foundation, particularly María Guerrero, Esme Plunkett, and Paúl Mayorga. Assistance and advice was also provided by: Jorge Carrión, Christian Sevilla, Danny Rueda, Jeffreys Málaga, Milton Chugcho, Rafael Chango, Jibson Valle, Francisco Calva, Edie Rosero and Francisco Moreno from DPNG. Novarino Castillo provided valuable field assistance. Institutional support was provided by DPNG (Dirección del Parque Nacional Galápagos), ECOGAL (Aeropuerto Ecológico de Baltra), FAE (Fuerza Aérea Ecuatoriana), ABG (Agencia de Regulación y Control de la Bioseguridad y Cuarentena para Galápagos), GAD (Gobiernos Autónomos Descentralizados from Floreana and Santa Cruz). Washington Tapia, Felipe Cruz †, María del Mar Trigo, and Frank Sulloway provided critical advice and encouragement. We also thank Washington Tapia and Frank Sulloway for their observations and data used to estimate age of maturity for Opuntias. Steve Rushton provided valuable comments on an earlier version of the manuscript. Finally, we thank our editor Timothy Scheibe and three reviewers: Neftali Ochoa-Alejo, Matthew Madsen, and F.B. Vincent Florens for their invaluable comments that improved this manuscript.

This publication is contribution number 2289 of the Charles Darwin Foundation for the Galapagos Islands.

### Funding

Funding was provided by the COmON Foundation, The Leona M and Harry B. Helmsley Charitable Trust, and the BESS Forest Club. There was no additional external funding received for this study. The funders had no role in study design, data collection and analysis, decision to publish, or preparation of the manuscript.

### Grant Disclosures

The following grant information was disclosed by the authors:
COmON Foundation, The Leona M and Harry B. Helmsley Charitable Trust.
BESS Forest Club.

### Competing Interests

The authors declare there are no competing interests.

### Author Contributions

- Patricia Isabela Tapia analyzed the data, authored or reviewed drafts of the paper, approved the final draft.
- Luka Negoita analyzed the data, prepared figures and/or tables, authored or reviewed drafts of the paper, approved the final draft.
- James P Gibbs conceived and designed the experiments, authored or reviewed drafts of the paper, approved the final draft.
- Patricia Jaramillo conceived and designed the experiments, performed the experiments, contributed reagents/materials/analysis tools, authored or reviewed drafts of the paper, approved the final draft.

### Field Study Permissions

The following information was supplied relating to field study approvals (i.e., approving body and any reference numbers):

Fieldwork and *Opuntia* plantings were approved by Dirección del Parque Nacional Galápagos (DPNG) under permit number PC-11-19.

### Data Availability

The raw data is available as a Supplemental File.

### Supplemental Information

Supplemental information for this article can be found online at http://dx.doi.org/10.7717/peerj.8156#supplemental-information.

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
