# Peer review of "Effectiveness of water-saving technologies during early stages of restoration of endemic Opuntia cacti in the Galápagos Islands, Ecuador"

_PeerJ, doi:10.7717/peerj.8156_

## Round 0.1 · original submission · Minor Revisions

The reviewers all find the study reported to be well designed and the paper generally well written. Each reviewer provides some specific comments that can be addressed through minor revisions or clarifications. Please provide an itemized response showing how each comment was addressed or providing an explanation if no changes were made.

·

Basic reporting

This article deals with an approach to try to increase the probability of survival of different Opuntia species by using two different water-saving systems (Groasis and Cocoon)and in four Galapago islands. The manuscript is well written and clearly organized. Literature references are sufficient and cover quite well the state of the art and also facilitate the discussion of results. In general, the manuscript is well structured and figures and tables are clear and well prepared. This article is technical and refers to the application of previous tested water-saving systems, but applied to the establishment of different Galapagos-threatened Opuntia species, whose populations have been diminished by different factors and need to be re-established in their natural ecological sites; this is the main importance and contribution of this article.

Experimental design

The experimental design of this work is right, and was focused to test two water-saving as an approach to try to increase the survival of different threatened Opuntia species in their ecological environments. The application of these systems is perhaps no quite original since they have been previously tested in some other plant species; however, their application as an alternative strategy to try to increase the survival of endangered or threatened Opuntia species is the important issue. This research was carried out with rigorous technical and scientific standards and the methods were clearly and sufficiently described to be replicated, if necessary.

Validity of the findings

The findings of this work were in some way contradictory when compared with previous reports cited by the authors since, in general, only partial success was attained by using the tested water-saving systems, and in some cases detrimental effects on the survival of Opuntia species were observed; however, the partial success can be applied to improve the rescue of Opuntia species in specific sites. In general, the results are robust, statistically sound and controlled. However, perhaps they should be used to get more information; for example, the authors mentioned in Table 2 that 349 individuals of Opuntia echios var. echios were planted in Baltra and 601 in Plaza Sur, and no any observation or comparation between the behavior or the survival of the same species in these two sites was mentioned. This should enrich the findings. Conclusions are well stated and limited to support the results.

Additional comments

The authors mentioned in Table 2 that 349 individuals of Opuntia echios var. echios were planted in Baltra and 601 in Plaza Sur, and no any observation or comparation between the behavior or the survival of the same species in these two sites was mentioned. Some minors corrections and observations are indicated in the text of the attached file.

·

Basic reporting

The manuscript by Tapia et al. is well written and should be published after minor editing. The introduction does a good job at providing background into why it is important to develop technologies to restore the endemic Opuntia cacti in the Galápagos archipelago, Ecuador. The authors also provide justification that the type of technologies should focus on improving water availability to the cacti. The Groasis Waterboxx and “Cocoon” are the technologies evaluated in the study to help improve water availability. Despite the importance of these technologies to the study, they are not mentioned in the introduction. I feel the paper would be improved by providing specific justification for testing these two technologies. Either in the introduction or the methods it would also be of value to explain in greater detail than what is in the paper how the two types of technologies work to improve water availability.

Experimental design

This research is original based off of the location it is being applied and the species of focus. The methods employed is also highly rigorous.

The study design is somewhat challenging to understand. The author’s state: “The number of controls was maintained at one control for every five Groasis or Cocoon technology treatment replicates.” Why was there not an even number of controls as the other treatments in the study? This should be clearly explained in the manuscript. Consider incorporating this information into the text or Table 1 and let the reader know how many replications of each treatment there is. I also do not understand how you randomized where you planted the different treatments. I assume the study utilized a completely random design? How the different treatments (control, Groasis, or Cocoon) were applied at different islands is also not clear in the methods section

Validity of the findings

The authors do a good job at explaining the results of their research and in describing the general impact this research has on the study of restoration ecology.

I feel the authors have correctly interpreted the results of their data and their conclusions are well stated.

Additional comments

Lines 138 -139 Please define what these terms mean “good,” “regular,” “poor,” and dead. Also, note you did not put quotation marks around “dead” like the other descriptive units.

Figure 4 caption does not provide an adequate description data being displayed. Please explain define what each of the categories mean

·

Basic reporting

The manuscript is generally well written but contains a few minor English mistakes and typos to be corrected (E.g. ‘parenthases’; ‘Opunitas’). The title of the manuscript may somewhat be currently overselling the findings in the sense that it seems “restoration” should really be replaced by something more like “early stages of establishment” (unless of course authors can clarify this point to satisfaction). On this topic, it seems also important for the authors to give the reader some context about the life history of the cacti species in nature, like for example, how long would it take for the life cycle to be completed? (from germination to seeding). This would give a better appreciation of the significance of the results in the broader scheme of things (how much more time would it take for the planted individuals to recruit into reproductive individuals, and therefore how close really are we to really achieving restoration of a reproductive population from the planted seedlings?).
The manuscript generally makes use of many references although certain specific statements lack reference support. E.g. line 73 should be backed by a reference as it is not automatically understood that active planting should necessarily be critical in the current case. There are however about 20 references that are listed in the reference section but which not find in the text. I suggest that authors consider using a bibliographic software, e.g. the freeware Zotero when preparing their reference section.
Abstract: About “restoring Opuntia population”: Is it really what is being achieved or is it rather “early stages of establishment of Opuntia” that should be written. How long does it take for a plant to mature and what is the longevity of these species?
Ok, I don't necessarily feel too strongly about it, but I feel that the number of display items used (five figures and two tables) is slightly on the high side and that the authors could consider moving one or two, of the least important ones, to supplementary materials.

Experimental design

No comment

Validity of the findings

No comment

Additional comments

How did human activities substantially reduce Opuntia population? Is it by direct destruction or changing the environment away from what is optimal for the Opuntia? For example, has human activities changed the soil parameters away from what is required by Opuntia? Why for example are the Opuntia not simply able to regenerate by themselves?

The authors mention that extreme aridity poses a major obstacle to Opuntia restoration, but are not these Opuntia precisely used to extreme aridity? Or do the authors mean that aridity has increased due to human activities, and that consequently it is now harder for Opuntia to regenerate in the wild? (Its mentioned that increased El Nino events pose a threat, but are these not making for wetter episodes rather than more arid ones?). I think it would be helpful to detail a bit why ‘extreme aridity’ poses such an obstacle to restoration, because if Opuntia are adapted to this, then it should be the wetter conditions posing a threat rather.

Minor comments:
L 34: Please specify the period over which this increased survival has been achieved, in the abstract.
L 70-71: The pronounced declines: Please can you specify at least roughly by how much and over which period, as these can be variously interpreted.
L 74: Please specify if such ‘severe aridity’ is something new (is it getting drier, and if so at what rate?) or whether it is just what the cacti are normally subjected to.
L 76: About ‘historically abundant’: It is a bit hard to figure out as it is a relative term. Can the author quantify at least roughly how much more abundant it was?
L 124: Can the authors briefly give the reader at least a rough appreciation of such declines? (e.g. halved in X years?)
L 134: Please specify whether this was done because only on those two islands, (among all those where the planting were done), had iguanas (or whether iguanas were also present on the other islands where the study was done).
L 125: ‘seedlings’ were planted rather than sown.
L 142: Please give the reader an idea of the extent of mortality in that age class, relative to later classes.
L 172: Write ‘seedling’ instead of ‘seedlings’
L 265-266: About ‘more than doubling the population’: Do you mean that the plants reached maturity? Or was it more just an increase in number of juveniles?
L 266: Any idea that could be given to the reader about what would have been the original natural densities? And what are the likelihood that these plants would eventually reach maturity (or are they already mature?)
L 295: What is the tune of these costs?

About the Cocoon and Groasis technologies: It is hard to visualize what these look like. Could the authors please provide some description, for example in supplementary material? This may be particularly relevant as size of cocoon is mentioned in discussion to explain the performance of plants treated with this technology.

I would suggest that ‘altitude’ be changed to ‘elevation’.

Referencing (in the body of the manuscript)
A) et al., should be italicised (= et al.,)
B) Line 80: Jaramillo, Cueva, Jiménez, & Ortiz, 2014 should be Jaramillo et al., 2014 [as cited later in the ms]
C) Line 133: Jaramillo et al., 2017 does not appear in reference section
D) Line 329: Jaramillo pers. obs. Should be P Jaramillo, date?, pers. obs.)

Reference section:
Please review, using the journal's style with a referencing software such as Zotero and Endnote would solve most problems found.
A) Spacing is not standardised
B) Add DOI (when available)
C) Section do not follow journal format of referencing: 1) Author initials should not have dots, 2) for 2 more authors, do not add & for the last one; 3) year should not be within parenthesis. For example Aldridge, C.L., Boyce, M.S. & Baydack, R.K. (2004). should be Aldridge, CL, Boyce, MS, Baydack, RK. 2004.
D) Journal name is not always italicised (line 389)
E) Article title should not have all words starting with capital letters like reference starting on line 404
F) Line 413: change var. Orientalis to var. orientalis
G) Line 471: change ‘(ed.).’ to ‘ed,’
H) Line 472: pp. 50-5. There is a missing number (50-55?)
I) Line 481 and 521: Remove italics from the publisher
E) Line 506: Second author is Li Z and not Zhiyang L
J) Line 506-507: Please italicise binomial.
K) Line 526: Remove volume and page as it is a software
L) Lines 406, 533 and 537: Add italics to journal

Many listed references are not cited in the text:
1) Aldridge, Boyce & Baydack (line 378)
2) Bensted-Smith (line 382)
3) Bisconti (line 384)
4) Campbell et al (line 396)
5) Cardinale et al (line 400)
6) Chapin et al (line 410)
7) Cruz et al 2005 (line 417)
8) Cruz et al 2009 (line 420)
9) Defaa et al (line 423)
10) Eckhard (line 432)
11) Ferwerda (line 437)
12) Ferwerda & Moolenaar (line 439)
13) González-Pérez, & Cubrero-Pardo (line 448)
14) Hamman (line 455)
15) Hanson & Campbell (line 457)
16) Helsen et al (line 459)
17) Lavoie et al (line 501)
18) Larson et al (line 503)
19) McDonald-Madden, Baxter & Possingham (line 511)
20) Poulakakis et al (line 525)
21) Snell, Stone & Snell (line 542)

Appendix (Opuntia_data.csv)
A) Add ‘n’ in the heading of column A.
B) Change ‘altitude’ to ‘elevation_m’ [column L]
C) Do you have such resolution to have elevation with two decimals?
D) Treatment cocoon from 2016-2018, treatment groasis and control 2013-2018

---

## Round 0.2 · accepted · Accept

Thank you for your careful attention to the reviewer comments. I agree that the revisions have significantly improved the manuscript and I recommend acceptance for publication.